# Factors Associated with Suicidal Behavior in Farmers: A Systematic Review

**DOI:** 10.3390/ijerph18126522

**Published:** 2021-06-17

**Authors:** Emelynne Gabrielly de Oliveira Santos, Paulo Roberto Queiroz, Aryelly Dayane da Silva Nunes, Kelly Graziani Giacchero Vedana, Isabelle Ribeiro Barbosa

**Affiliations:** 1Post-Graduate Program in Collective Health, Universidade Federal do Rio Grande do Norte (UFRN), Natal RN 50056-000, Brazil; enfpaulorq@gmail.com (P.R.Q.); aryellydayane@gmail.com (A.D.d.S.N.); isabelleribeiro68@gmail.com (I.R.B.); 2Post-Graduate Program in Psychiatric Nursing, Universidade de São Paulo (USP), São Paulo SP 14040-902, Brazil; kelly.giacchero@gmail.com

**Keywords:** suicide, epidemiological factors, farmers, mental health

## Abstract

This review aimed to investigate the factors associated with suicidal behavior in farmers in the scientific literature. Two researchers participated independently in searching databases, specifically PubMed/MEDLINE, LILACS, Web of Science, Scopus, PsycINFO, and SciELO. Only observational studies were included. The quality of the selected studies was assessed with a critical assessment checklist for cross-sectional analytical and case-control studies, prepared by the Joanna Briggs Institute. Data related to the publication were collected (author and year; city/country); methodological design; sample/population (gender; average age), outcome, measuring instrument and factors associated with suicidal behavior. A total of 14 studies were included in the systematic review, and factors associated with farmers’ behavior in mental health (depression), seasonal impacts (drought), and work exposures (herbicides and insecticides) were identified. However, heterogeneity was found in terms of the method, measurement of suicidal behavior, and associated factors, which indicates the need for further studies.

## 1. Introduction

Living and working conditions in the countryside can hide the harsh reality of the situations of poverty, unemployment, precarious access to health and education, and also situations of exploitation, violence, fear, insecurity, and the presence of common mental disorders, constituting some risk factors for suicidal behavior in rural populations, especially in farmers [1,2,3,4].

According to the World Health Organization (WHO), suicidal behavior is associated with different outcomes and actions with varying degrees of lethality, such as suicidal ideation, attempted suicide, and completed suicide [5,6]. It is important to consider that this phenomenon encompasses, regardless of the point of view through which it is analyzed, a central dimension related to distress, and the individual understanding of each of these outcomes is related to a better understanding of risk factors, which can direct more effective prevention strategies [7,8]. The scientific literature evaluates these different phenomena using different resources.

Studies have shown that inequalities in suicide rates have been found in the rural or urban areas, with the highest rates in the rural area and, above all, in farmers [6,9]. In India, for example, countless farmers have taken their lives since the 1990s, which raised the suicide rate from 12.3 in 1996 to 19.2 in 2004 [10]. In Canada, agriculture is one of the occupations most at risk of suicide death, with a mortality rate of 31.4 suicides for every 100,000 individuals [11]. Although largely preventable, the proper management of individuals with suicidal behavior is a challenge for health systems and services worldwide due to the high cost and complexity of the phenomenon.

This is a systematic review of 65 studies, in which we included 32 of them in the meta-analysis, quantified the risk of suicide in farmers, identifying an effect size of 1.48, representing an excess risk of suicide in this population, varying according to the geographic area, with the highest risk observed in Japan [12]. Another systematic review that identified the risk factors that affect farmers’ mental health also showed that 71% of the included studies suggested that farmers have worse mental health problems than the general population, constituting an important trigger for suicidal behavior [13].

Some researchers have focused their attention on studying suicidal behavior in the rural population by studies conducted with farmers, specifically [14]. However, the aspects inherent to this phenomenon in this group need to be better evaluated, given that the social, economic, life, and work aspects, common to farmers, can explain the factors associated with suicide and its components, and may differ from those observed only in rural populations [15].

Considering that the existing literature offers little guidance on how factors associated with suicidal behavior in farmers may differ from those in rural communities more widely, a refined search and evaluation in the literature is essential for the definition of health actions and policies aimed at the prevention of this disease, early interventions and promotion, surveillance, and health care, considering the most relevant target factors in this group.

It is also relevant to investigate patterns of association of these risk factors in the world and whether there are variations according to the region studied. Thus, based on the results described and the gaps presented in those studies, this study aimed to conduct a systematic review of the scientific literature on the factors associated with suicidal behavior in farmers.

## 2. Materials and Methods

The study of systematic literature review on factors associated with suicidal behavior in farmers followed the recommendations of the PRISMA Check-list (Preferred Reporting Items for Systematic Reviews and Meta-Analyzes), for its report and the construction of the protocol, registered in the PROSPERO (CRD42020164947) [16]. The focus question of the research was “What factors are associated with suicidal behavior in farmers?”.

### 2.1. Eligibility Criteria

The review included observational scientific articles with a quantitative approach that investigated factors associated with suicidal behavior in farmers, without restrictions on language, place, and period of publication. We excluded investigations with qualitative methodology, reviews, theoretical essays, research protocols, methodological articles, and those who did not present risk and/or association measures, such as Odds Ratio, Relative Risk, and Correlation Coefficients. Moreover, studies with a rural population that did not include farmers as an object of analysis were excluded. All individuals were included in the study, with no age or gender limits.

### 2.2. Literature Search

The steps of the review were held independently by two researchers (E.G.O.S and P.R.Q). The researchers consulted the electronic databases PubMed/Medline, Lilacs, Web of Science, Scopus, PsycINFO, and Cinahl. In the gray literature, Google Scholar, ProQuest, and OpenGrey were consulted. The main descriptors related to the investigated theme were crossed: farmer and suicide. All electronic searches were performed on 28 March 2021.

### 2.3. Selection of Studies

Initially, we inserted all selected articles and removed the duplicates in Mendeley. Subsequently, we read titles and abstracts in Rayyan QRCI, excluding all those who did not meet the eligibility criteria. We read the selected studies in full and excluded those that did not meet the eligibility criteria. When there were conflicts, a consensus meeting was held and, when necessary, the third reviewer (I.R.B) was consulted [17].

### 2.4. Data Collection Process


While reading the articles included in the review, the researchers individually and blindly completed the stage without viewing the respective findings of the other, extracting data regarding the author and year of publication, city/country in that the study was carried out, design, sample/population, gender, average age, outcome related to suicidal behavior (suicide, suicidal ideation or attempted suicide), instrument used for outcome analysis, the measure of association, and factors associated with suicidal behavior. Subsequently, there was a consensus meeting to compare the included data to minimize possible inconsistencies in the interpretation of the extracted data.

### 2.5. Bias Risk Assessment

The Critical Evaluation Checklist for cross-sectional analytical, cohort, and case-control studies assessed the quality of the selected studies, prepared by the Joanna Briggs Institute. The results of the methodological quality for each design and figures were elaborated in the Review Manager 5.3 (RevMan 5.3) [18].

The reviewers classified the articles into three levels: (1) low risk of bias (if the studies reached a “yes” score in at least 70% of the items evaluated; (2) moderate risk of bias if the “yes” score was between 50% and 69%; and (3) high risk of bias if the “yes” score was less than 49% [19].

## 3. Results

### 3.1. Selection of Studies

We identified 4113 articles that addressed the factors associated with suicidal behavior in farmers. We read in full 66 articles and, after applying the eligibility criteria, 14 studies were the object of this analysis (Figure 1), whose characteristics of these included studies will be presented in Table 1.

The major cause for exclusion were the studies that did not present the outcome of associated factors in the farmer population, although they were conducted in this population. The authors of the 11 studies that did not present the association measures were contacted to obtain this information through the e-mails provided in the publication, but no response was obtained.

Regarding the geographic distribution and population of the analyzed studies, there was heterogeneity in the results. The studies evaluated different age groups, and only 21% of them analyzed the average age of the population, which was approximately 52 years [15,21,28]. Moreover, 11 studies presented the variable age categorically. The majority of the study population was composed of men and women, and 21% of them studied only men (Table 1).

Regarding suicidal behavior, eight studies had suicide as an object of analysis [15,20,21,22,24,27,29,31], and the others analyzed attempted suicides [26] and suicidal ideation [25,28,30,32]. Different measures for identifying associated factors were also observed, with bivariate analysis (OR/RR) [15,20,22,23,25,26,27,29,30] and correlation coefficients (r, β, α) [15,24,28,31,32], and differences in the types of associated factor analysis techniques.

The factors associated with suicidal behavior were heterogeneous, and ranged from aspects proximal to the individual (advanced age, food insecurity, depression, psychiatric illnesses, stress and mental distress) to aspects inherent to the activity in the field (indebtedness and pesticide use); the instruments used to check the outcome were also heterogeneous among the studies. 

While McLaren S. and Chantal C. [28] used the General Health Questionnaire, Sweetland AC, et al. [32], used the PRIME-MD, and Bjornestad A, Curthbertson C, and Hendricks J [21] used the SBQ-R. The other authors used semi-structured questionnaires containing a guiding question about suicidal behavior [20,23,25,26,30]. In the study by Picket, et al. [29], which emerged from a cohort, the variables were extracted and analyzed from databases, without presenting the instrument used for collection. Accordingly, knowing that the included studies presented different analysis techniques, instruments, and association measures, characterizing a heterogeneity in the results, it became unfeasible to proceed with a meta-analysis of the results in this review study.

### 3.2. Risk of Bias in the Studies

Regarding risk of bias, in the cross-sectional studies, a score of ≥71.4% “yes” answers was observed in most studies, thus demonstrating low risk of bias. The main weakness was related to the fact that the authors did not clearly report any method to deal with the confounding factors. Therefore, these studies were considered high risk of bias only with regard to this item. On the other hand, the study by Hanigan, et al. [22], observed a high risk of bias, with only 42% of “yes” answers, as can be seen in Figure 2 [21,22,23,24,30,31,32].

For the case-control studies, although in an overall analysis a low risk of bias (80%) was observed, Pickett et al. [29]. did not clearly define whether exposure was measured in the same way for cases and controls, as well as whether the exposure period of interest was long enough to be meaningful. In turn, Bhise and Behere [20] did not present the confounding factors and the strategies adopted to deal with these factors (Figure 3).

For the retrospective cohort [15], a low risk of bias was observed, since although the authors listed the confounding factors in the study, they did not make clear the strategies for dealing with these factors (Figure 4).

## 4. Discussion

The present systematic review sought to identify, through the literature, the factors associated with suicidal behavior in farmers. Among the 14 identified studies, the factors were mainly related to mental health (depression), seasonal impacts (drought), and work (pesticides use and indebtedness). It was possible to observe a significant variation in the method of the studies, investigated factors, ways of measuring the outcome, and investigated population, which resulted in a wide variability of associated factors that, in general, were related to individual aspects, mental health, and agricultural activity.

It is important to emphasize that most studies were excluded during the selection stage of this systematic review because they were conducted with rural populations in general, and did not evaluate factors associated with suicidal behavior in farmers, the object of study of this analysis. Moreover, those studies that did not present the association measures in the results were also excluded. It is also noteworthy that contact was established with the authors of these studies in order to obtain the completeness of the analyses of the results through the association measures for the factors associated with suicidal behavior in farmers.

Thus, in order to better understand the factors related to suicidal behavior in farmers elucidated in the literature, it is important to highlight that variables such as age, gender, and places where the studies were conducted constitute important elements in the understanding of the analyzed outcome, especially when taking into account that they can also be related to the outcomes. It becomes, therefore, an important element to be presented not only in the characterization of the sample, but also as a variable. Nevertheless, in this systematic review, only one study reported an association of the outcome with age, by showing a correlation between advanced age and suicidal ideation present in farmers [32]. 

The significant impact of suicide mortality in the elderly population is observed worldwide, although even in this population non-lethal suicidal behavior may have started previously. In Brazil, for example, Santos and Ribeiro [33] found that the suicide mortality rate for this group showed a statistically significant trend of increase between 2000 and 2014, for both men and women. There are several hypotheses and explanatory models to explain the increased risk of suicide in the elderly population. The prominent and characteristic risk factors of this group are the loss of social roles, functional impairment and loss of autonomy, greater lethality of attempts, lack of adaptation to changes, multiple losses, hopelessness, difficulty in exercising control over important aspects of life, and frustrated belonging [34,35,36,37,38].

Moreover, a large part of the studies in this review analyzed the inherent aspects of suicidal behavior for both genders [15,20,21,22,23,26,27,31,32]. In different countries, suicide rates are higher in men, while non-lethal methods, such as suicidal ideation and attempted suicides, in women. Since suicidal behavior is analyzed in this review and it is not limited to lethal methods, it is important to highlight that men use more violent methods when attempting suicide [39,40].

Regarding the study site, most studies (71%) were conducted in developed countries, and did not investigate regional variables [15,21,22,23,24,25,28,29,30,31]. High-income countries have higher suicide mortality rates, although the most suicide deaths occur in low- and middle-income countries (79%), with a greater concentration of the world population (84%) [41]. Historically, most studies investigating this topic are in Europe and other equally wealthy countries such as the United States and Japan. Little is known about the influence of certain factors such as those inherent in agricultural activities, in suicide rates in South America, and especially in Brazil, a country known to be unequal and with intense agricultural activity [37].

The factors associated with suicidal behavior in farmers may differ from one another from the analysis of the studied outcome: suicidal ideation, suicide attempt or suicide [8]. In this sense, the importance of understanding each of these factors, from the outcomes, in their particularities, stems mainly from the fact of directing the planning of prevention strategies of the phenomenon more effectively in this population. 


The figure below illustrates the associated factors identified in the 14 studies included in this review (Figure 5). Furthermore, it presents the proportional calculation of these factors from the main observed categories: work, mental health, and social and individual aspects. In this case, a higher percentage of factors associated with work were observed. Thus, it is important to highlight the necessity of an integral perspective of care with regard to workers’ health, especially when considering the impact of the agricultural work regarding the suicidal behavior of this population.

### 4.1. Aspects of Work

Among the associated factors found, some studies have noted that indebtedness is related to farmer suicide, especially when reporting active farm debt rates, and especially in the last 5 years [20,31]. Small farms, for example, maybe more vulnerable to financial pressures and market fluctuations. Thus, it can culminate in the indebtedness of the farmer, especially from bank loans or debts with family and friends. Farmers inserted in the scenario of small-scale farms may not be able to bear the serious consequences of reduced agricultural production since they have detrimental effects on their family’s financial situation. This situation together with a fragile social context can lead the individual to psychic disorders such as anxiety, depression, and, as a consequence, suicidal behavior [43].

Furthermore, the impact of drought on agricultural activity has also been shown to be associated with suicide, reported in the study by Hanigan, Butler, Kokic, and Hutchinson [22], where it was found that an increase in drought index increases the risk of suicide by 15%, as does an increase in monthly maximum temperature, where this risk increases by approximately 3% (*p* < 0.001). Studies conducted in India, where more than half of the population depends on agriculture, corroborate these findings; they also showed that most participants expressed the significant negative impact of drought on them, believing they would not be able to recover from the negative consequences caused. In addition, it was observed that suicide risk was inversely correlated to water availability and groundwater monitoring, rapid emergency measures, as well as monitoring of farmers for suicide prevention in higher risk areas are recommended [44,45]. A literature review study found approximately 20 articles with reports of the effects of drought on the mental and emotional health of public health, with elderly farmers in rural areas as the most studied groups [46], thus highlighting the need for a globalized look at the farmer’s health.

Another factor associated with suicidal behavior, elucidated in this review, is about the aspects inherent to employment relationships. This is because there was an association between suicide and seasonal agricultural work (a form of temporary work associated with periods of the year and specific sectors) when compared to work with longer contracts [29,46].

In agricultural production, seasonality is mainly due to the climatic variations of the seasons that affect planting times, crop development, and harvest [47,48]. In the context of agriculture, the seasonality of work implies precariousness and lack of stability, which can be a risk factor for greater psychological distress in the farmer [49], thus contributing to suicidal behavior. On the other hand, annual work is also associated with other risks such as the time of exposure to pesticides.

There was also an association with regular employment relationships in which work is paid for a longer period of time [29]. The rural environment has undergone transformations and is no longer essentially agricultural, potentially representing a place of residence. Small farmers are increasingly integrated into a larger system and articulation with industry and non-agricultural activities have been gaining space among residents of rural areas. As non-agricultural activities are alternatives to income generation, they undermine the farmer’s identity, impoverishing his culture and technique [50].

Pesticides, as chemicals widely used in agriculture, have caused immeasurable damage to human health, increasing the severity in cases of intoxication. They are used in native and planted forests, in water, urban and industrial environments, and, to a large extent, in agriculture and pastures for livestock [51,52]. When considering agriculture as an active setting for pesticide use, studies have shown the relationship between suicidal behavior and this practice.

An increased chance of suicidal ideation has been observed from poisoning, either during one (OR = 2.33) or more than one (OR = 3.02) episodes, as well as an approximately threefold increased chance of suicidal ideation in cases of hospitalizations for pesticide poisoning. The more severe the symptoms related to pesticide poisoning, the greater the chances [25]. Furthermore, in Canada, it was also observed in a study with farmers that pesticide spraying doubles the chances of suicide (OR = 1.71) [29]. In Brazil, cities with tobacco production were also associated with suicide cases among farmers (OR = 2.39) [27].

Studies found risk for suicidal behavior in people who handled pesticides, and that long exposure to these products leaves neurobehavioral sequelae, which can progress to depression [53]. This data combined with a range of social and economic problems could cause a suicide [54,55]. Exposure to cholinesterase-reducing agents can cause behavioral changes that can lead to suicide in depressed or anxious people. Many pesticides in common use in the agricultural sector result in low levels of cholinesterase in exposed people [29], thus demonstrating the cause of the relationship between pesticide use and suicidal ideation in this audience. 

However, when determining suicide mortality in the population of a Brazilian municipality, Lima [55] sought to verify the association between exposure to pesticides and suicide. The results of the study showed that there is no statistical difference in the practice of suicide in workers who use pesticides.

### 4.2. Aspects of Mental Health

On the other hand, there are aspects related to farmers’ mental health that are associated with suicidal behavior. In this review, it was observed that mental distress and depression were associated with suicidal ideation [28,30,32]; and that the presence of psychiatric illness, the presence of feelings of self-blame, and the farmer’s day-to-day stress increased the chances of suicide in farmers [20].

Isolation, issues linked to loss of health, and the impossibility of exercising daily activities are issues that need to be considered, although we did not investigate them in this study. In this sense, the rural area has been occupying a precarious place, which implies the loss of social objects, both real and idealized ones: there is a loss of health, work, social status, importance in the family nucleus, financial losses, family bonds, and social relationships and affective bonds. As a result, the farmers’ mental health is progressively damaged, there are signs of difficulties and impediments to living, and suicidal behavior is gaining space in the farmer’s life [56].

Among the various explanations for these findings, the stressful events to which farmers are subjected, especially in the occupational environment, deserve to be highlighted. Many of them result from the transformations that have occurred in the agricultural process since the replacement of traditional forms of agriculture by the aggressive penetration of agribusiness, its technological packages of mechanization and intensive use of pesticides, in addition to economic factors. Moreover, mental illness in the agricultural environment can be subject to stigma and misunderstanding, which hinders the search for health, often worsened by geographical isolation and difficulty in accessing health services [57].

A study conducted in Brazil identified a range of acute and mental health symptoms in farmers and their assistants, such as headache and depressive signs. Possible findings to explain part of these symptoms were related to the fact that they use fewer items of personal protection during the use of pesticides. In light of this, the need for surveillance actions, technical support, and safety training during occupational and environmental exposure to pesticides should be encouraged as a way to impact on the reduction of these causes, in order to decrease the outcomes related to suicidal behavior in farmers [58].

### 4.3. Individual and Social Aspects

In addition to aspects related to agriculture and farmers’ mental health, it was also identified that individual aspects related to socioeconomic context, such as low socioeconomic status; living in areas with high levels of deprivation and with family with poorer assets; primary education level; and households with self-reported alcohol problems, are related to higher chances of attempted suicides in farmers.

Although the associated factors mentioned above are related to suicide attempts, it should be considered that they can also contribute to the risk of suicide as an accomplished act, especially in the absence of effective prevention strategies. Nonetheless, these results are discordant when compared to a study conducted in Brazil, which showed no spatial autocorrelation between socioeconomic variables and suicide mortality rates [Santos]. In general, some studies support the hypothesis that social inequality increases suicide rates [59].

In this review, factors such as employability are also associated with suicidal behavior. One of the studies showed that having a job increases the chances of suicide in farmers by almost two times [15]. Conversely, in some countries, such as India, it is observed that the highest suicide rates, especially in men, are present in states with higher levels of unemployment [60]. 

For Durkheim, society also plays a key role in the construction of the individual. Therefore, social factors, such as family, school, the groups in which he/she participates, friends, and society, have a significant influence in the production of a suicidal episode, both for it to occur and to be avoided [61]. These implications can also be understood when considering factors associated with suicidal ideation from an individual perspective, such as impulsivity, domestic violence, and aggression. In this study, in particular, it was observed that these factors were associated with both suicidal ideation and attempted suicides in farmers [30].

Accordingly, strategies for coping with suicidal behavior in farmers can be used, such as a monitoring and support system for vulnerable farmers, support and counseling services, and training among farmers. Therefore, we need to strengthen, at national and international levels, public health programs and policies related to mental health, work, and economic support, especially to small farmers to leverage the social promotion of rural workers, and bring to the families involved economic resources capable of guaranteeing their social reproduction [62].

We had some limitations in this review such as the methodological differences for assessing the outcomes related to suicidal behavior and estimation of the associated factors in the included studies, especially when considering the absence of specific and validated instruments for assessing suicidal behavior. This methodological heterogeneity, observed in both the instruments and the associated factors, shows the need to carry out new primary studies related to the topic since suicidal behavior is considered a public health problem, and vulnerable groups, such as farmers, need early interventions to avoid fatal outcomes and improve the mental health and quality of life of this population. However, the analysis by two researchers independently, the evaluation of the quality of the studies, and the resolution of discrepancies by agreement sought to reduce possible biases.

The limitations found in most of the included studies were mainly related to the absence of methods for identifying confounding factors and confounding adjustments. Few studies reported the potentially important confounding factors that were selected from the available factors [15]. It is also noteworthy that all included studies underwent methodological quality assessment and were judged as having low risk of bias, since, when analyzing the aspects inherent to the critical assessment list of studies, they obtained scores >70%, except for one cross-sectional study that showed high risk of bias [22].

## 5. Conclusions

This review identified factors associated with behavior in farmers related to individual aspects, mental health, seasonal and work impacts, such as pesticide use and indebtedness. In general, it was noted that attempted suicides were related mainly to socioeconomic factors; and suicidal ideation and completed suicide to mental health and agricultural aspects, such as the use of pesticides. Thus, this study shows guidelines for research on the theme and reinforces the importance of investigating the strength of association between variables inherent in the dynamics and work in the area of farmers, and the prevalence of different forms of presenting suicidal behavior to better understand the phenomenon and direct more effective public health policies to prevent this problem in farmers.

## Figures and Tables

**Figure 1 ijerph-18-06522-f001:**
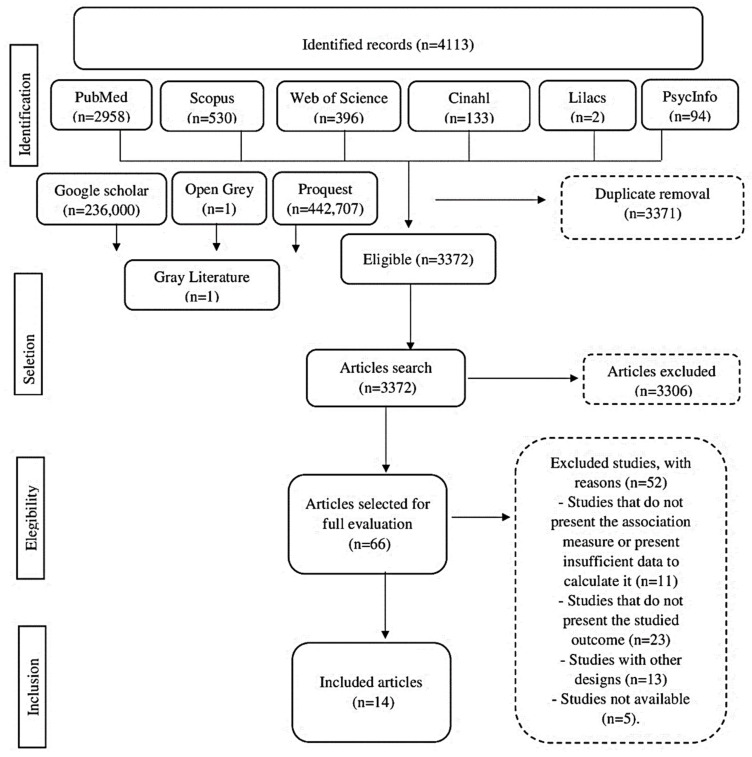
Flowchart of the selection of articles included in the review.

**Figure 2 ijerph-18-06522-f002:**
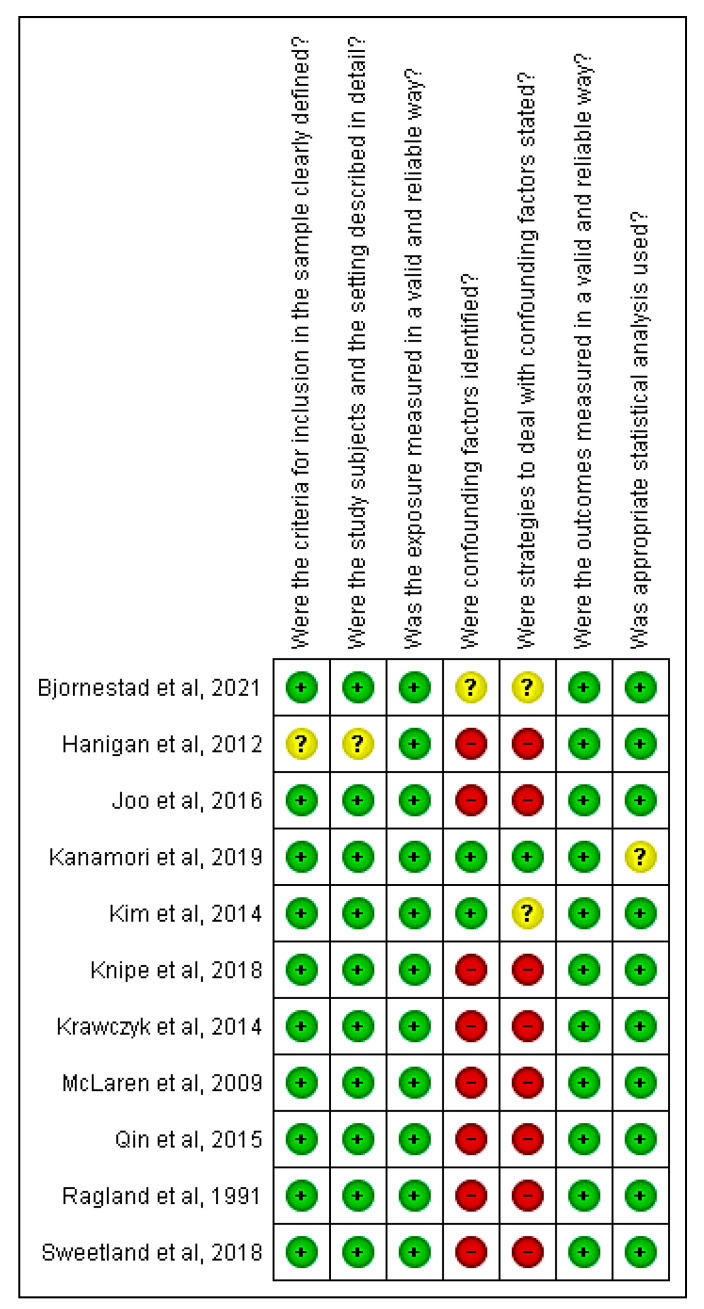
Bias risk for cross-sectional studies included in the review.

**Figure 3 ijerph-18-06522-f003:**
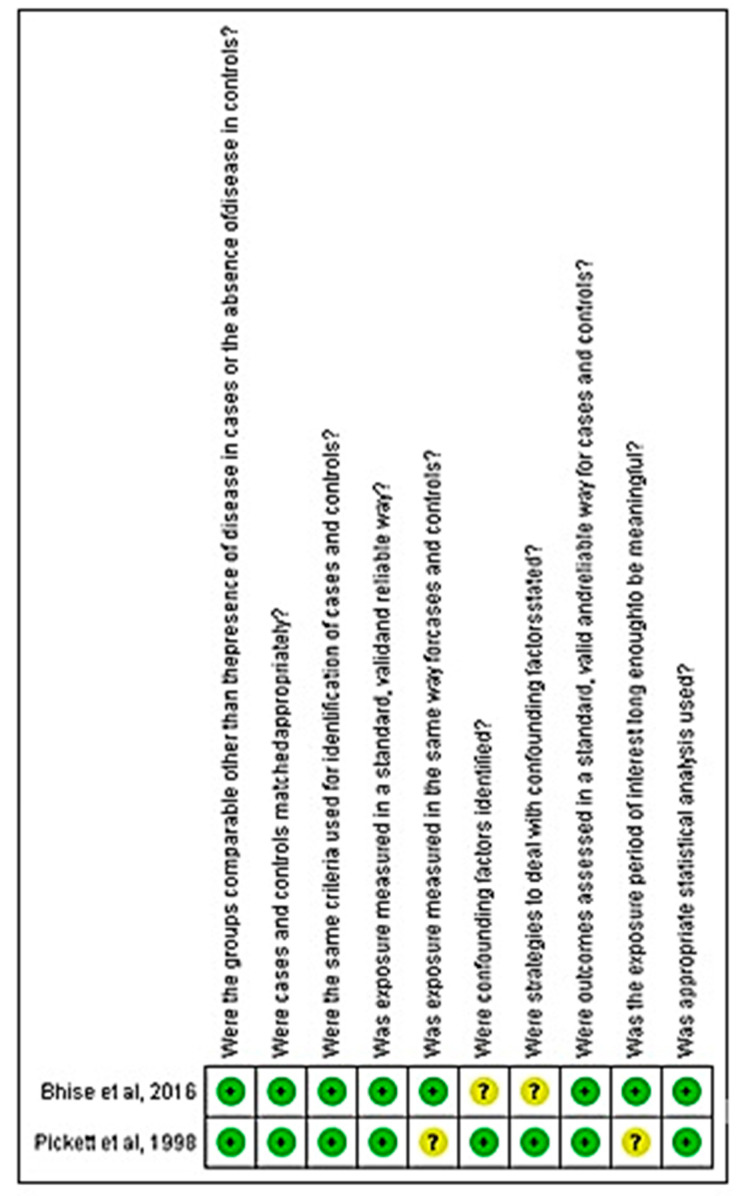
Analysis of the risk of bias for the case-control study included in the review.

**Figure 4 ijerph-18-06522-f004:**
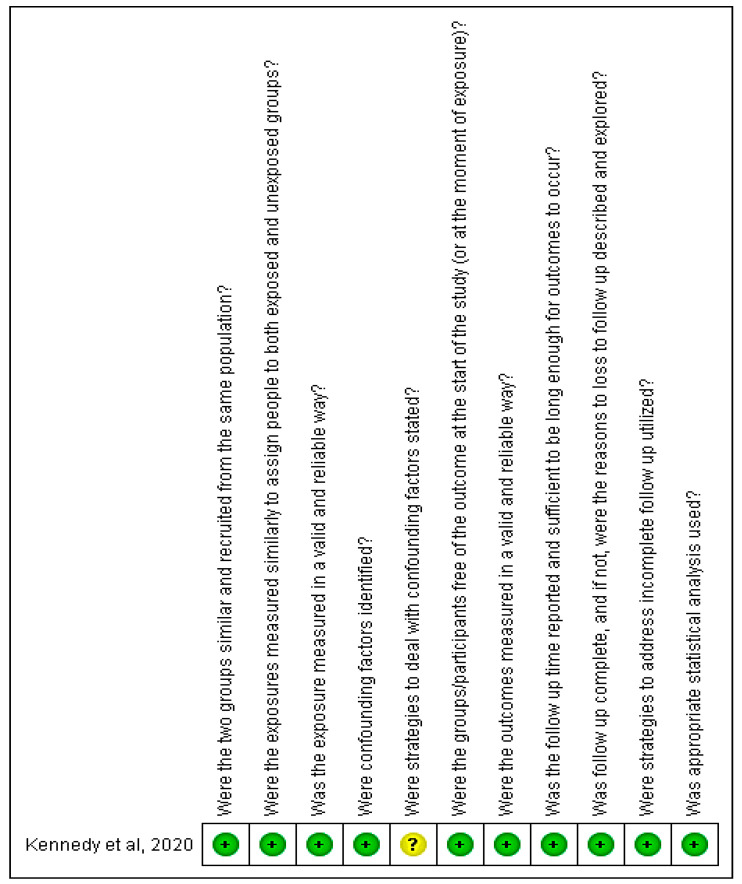
Analysis of the risk of bias for the cohort study included in the review.

**Figure 5 ijerph-18-06522-f005:**
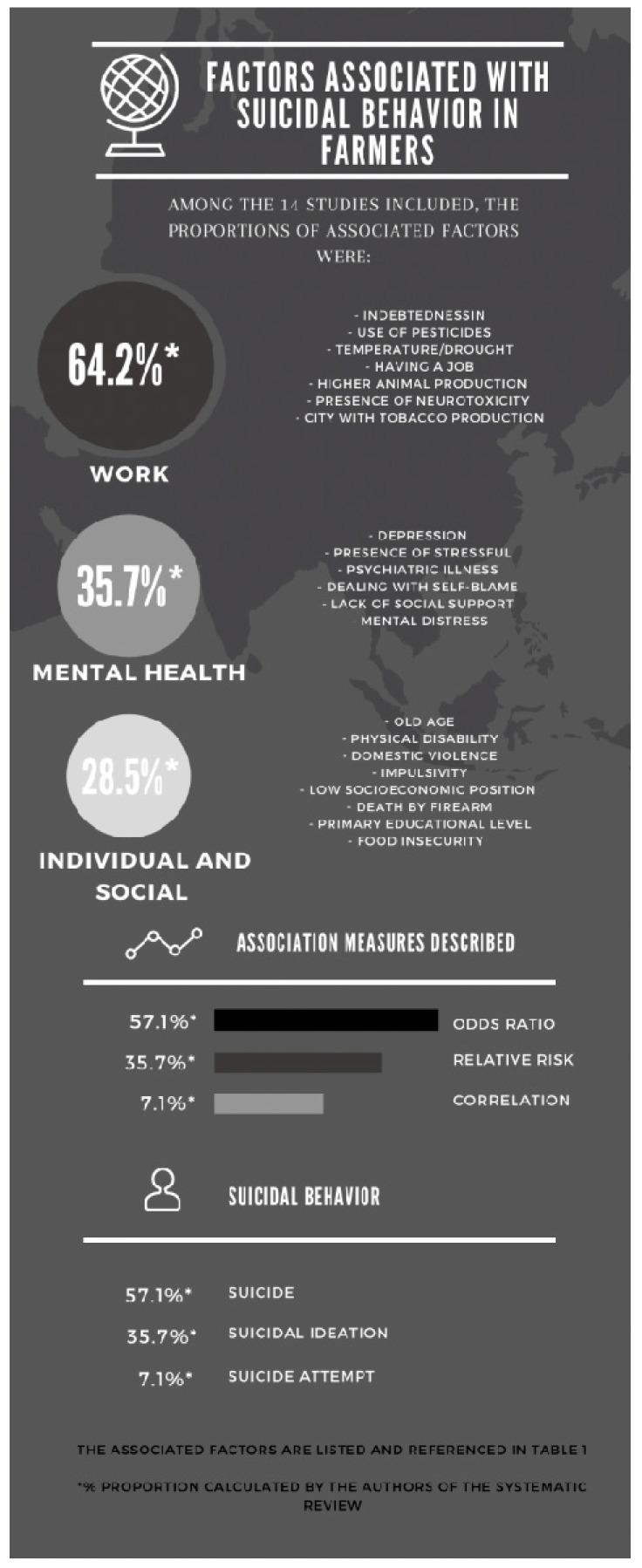
Factors associated with suicidal behavior in farmers. (App used: Canva Pty Ltd. [42]).

**Table 1 ijerph-18-06522-t001:** Characteristics of included studies.

Author and Year of Publication	City/Country	Study Desing	Sample	Gender	Age (Average ± Standard Deviation)	Outcome	Measuring Instrument	Factors Associated with Suicidal Behavior
Bhise MC, Behere PB; 2016 [20]	Vidarbha/Maharashtra/India	Case-control	Cases: 98 Controls: 98	Both genders	-	Suicide	Semi-structured questionnaire	Indebtedness in the past 5 years (OR = 3.86); presence of diagnosable psychiatric illness (OR = 7.81); presence of stressful life events in the past year (OR = 3.20)
Bjornestad A, Curthbertson C, Hendricks J; 2021 [21]	EUA	Cross-sectional	600	Both genders	63 ± 12.5	Suicide	SBQ-R	Dealing with self-blame (β = 0.065; *p* < 0.05; increase in risk by 6.7%)
Hanigan IC, Butler CD, Kokic PN, Hutchinson MF; 2012 [22]	New South Wales/Australia	Cross-sectional	-	Both genders	-	Suicide	-	Increase in monthly maximum temperature (RR = ±3%, *p* < 0.001); increase in drought index (RR = 15%, *p* < 0.001)
Joo Y, Roh S; 2016 [23]	South Korea	Cross-sectional	543	Both genders	-	Suicidal ideation	Semi-structured questionnaire	Lack of social support (OR = 2.13); working 5–8 h per day (OR = 2.45); presence of neurotoxicity (OR = 6.17); farmer’s syndrome (OR = 3.70)
Kanamori M, Kondo N; 2019 [24]	Japan	Cross-sectional	-	Both genders	-	Suicide	-	Higher animal production per population unit (β = 8.46)
Kennedy A, Adams J, Dwyer J, Rahman MA, Brumby S; 2020 [15]	Australia	Cohort retrospective	1298	Both genders	47	Suicide	-	Having a job (OR = 1.84); death by firearm (OR = 4.51) *p* < 0.001
Kim J, Shin DH, Lee WJ; 2014 [25]	South Korea	Cross-sectional	1895	Male	-	Suicidal ideation	Semi-structured questionnaire	Hospitalization for pesticide poisoning (OR = 2.48); pesticide poisoning (OR = 2.33 for once; OR = 3.02 for more than once); severity of moderate or severe symptoms from acute pesticide poisoning cases (OR = 2.23)
Knipe DW, et al.; 2018 [26]	Sri Lanka/India	Cross-sectional	165,233	Both genders	-	Suicide attempt	Semi-structured questionnaire	Living in a household with poorer assets (OR = 2.37); low socioeconomic status (OR =1.45); living in areas with a high percentage of households with a self-reported alcohol problem (OR = 1.44); primary education level (OR = 3.27)
Krawczyk N, Meyer A, Fonseca M, Lima J; 2014 [27]	Alagoas/Brazil	Cross-sectional	122,036	Both genders	-	Suicide		City with tobacco production (OR = 2.39)
McLaren S, Chantal C; 2009 [28]	Victoria and New South Wales/Australia	Cross-sectional	99	Male	48.14 ± 12.04	Suicidal ideation	General Health Questionnaire	Depression (r = 0.55; *p* < 0.001)
Pickett, et al., 1998 [29]	Canada	Case-control	Cases: 1457 Controls: 11,632	Male	-	Suicide	-	-
Qin Q, Jin Y, Zhan S, Yu X; 2015 [30]	China	Cross-sectional	939	Female	-	Suicidal ideation	Semi-structured questionnaire	Herbicide and insecticide spraying (OR = 1.71); seasonal versus year-round farm work (OR = 1.68); high levels of paid labor (OR = 1.61); physical disability (OR = 7.43); domestic violence (OR = 2.65); depression (OR = 1.07), impulsivity (OR = 1.04) and motor impulsivity (OR= 1.07)
Ragland JD, Berman AL; 1991 [31]	EUA	Cross-sectional	-	Both genders	-	Suicide	-	Active farm debt rates (* r = 0.36, *p* < 0.0028)
Sweetland AC, et al., 2018 [32]	Nigeria, Uganda and Ghana/Africa	Cross-sectional	1142	Both genders	-	Suicidal ideation	PRIME-MD	Mental distress (Nigeria β = 0.731, *p* < 0.001; Uganda β = 0.584, *p* < 0.001; and Ghana β = 0.350, *p* < 0.001); food insecurity (Nigeria β = −0.255, *p* < 0.05); old age (Ghana β = 0.218, *p* < 0.05).

OR = Odds Ratio; r = correlation coefficient; β = coefficient of association in the logistic regression model; * Statistical significance: *p* < 0.05/Multiple Model: *p* < 0.001.

## Data Availability

Not applicable.

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
