# Peer review of "Factors Associated with Suicidal Behavior in Farmers: A Systematic Review"

_ijerph, 2021, doi:10.3390/ijerph18126522_

Round 1
Reviewer 1 Report
23 April 2021
The 3nd Review on the manuscript titled “Factors associated with suicidal behavior in farmers: a systematic review” by Santos EGO. et al, submitted to IJERPH.
Manuscript ID: ijerph-1119339, ijerph-1212488
Dear Authors,
The authors systematically reviewed the scientific literature regarding factors associated with suicide in farmers. Out of six databases, two independent researchers found 4113 articles, of which fourteen studies met the final criteria. The authors concluded that several factors were associated with suicide such as depression, drought, herbicides, and insecticides, emphasizing on the importance of further studies regarding factors inherent in farmers, which lead to suicidal behavior.
Please consider following:
- Page 3: “performed on April 28, 2021”; The search date cannot be after the date of resubmission.
- Pages 7-10, Discussion: Please include a figure to present the significance of the study. It can be like a graphical abstract.
The manuscript contains four figures, one table and 63 references. The quality of the manuscript was substantially improved. More references are expected. Suggested reference: Liang, Y.; Wang, K.; Janssen, B.; Casteel, C.; Nonnenmann, M.; Rohlman, D.S. Examination of Symptoms of Depression among Cooperative Dairy Farmers. Int. J. Environ. Res. Public Health 2021, 18, 3657. The manuscript carries important value regarding the factors and future tasks associated with suicidal behavior. I recommend this manuscript for publication after minor revision.
I declare no conflict of interest regarding this manuscript.
Best regards,
Author Response
Natal (RN) April 28, 2021.
Dear Editor and Reviewer,
Journal: International Journal of Environmental Research and Public Health
We would like to thank you for the opportunity for our manuscript entitled “Factors associated with suicidal behavior in farmers: a systematic review” to be evaluated by the journal and for the valuable contributions of reviewers to improve the work. We send below the comments of the reviewers with the answers, and the artile with the adjustments.
We are at your disposal for any clarifications and/or other necessary adjustments.
- The search date (28 march 2021) was corrected in the manuscript, in the methods section.
- We do not include the indicated reference (Liang et al, 2021). The study does not answer the question in this review, which is "what factors are associated with suicidal behavior in farmers?"
- The figure was included in the discussion (Figure 2). The figure illustrates the associated factors identified in the 14 studies included in this review. Furthermore, it presentsthe proportional calculation of these factors from the main observed categories: work, mental health and social and individual aspects.
Reviewer 2 Report
The paper is really improved!
Congratulations!
I recommend publication of the paper as is.
Author Response
Natal (RN) April 28, 2021.
Dear Editor and Reviewer,
Journal: International Journal of Environmental Research and Public Health
We would like to thank you for the opportunity for our manuscript entitled “Factors associated with suicidal behavior in farmers: a systematic review” to be evaluated by the journal and for the valuable contributions of reviewers to improve the work. We send below the comments of the reviewers with the answers, and the artile with the adjustments.
We are at your disposal for any clarifications and/or other necessary adjustments.
- The search date (28 march 2021) was corrected in the manuscript, in the methods section.
- We do not include the indicated reference (Liang et al, 2021). The study does not answer the question in this review, which is "what factors are associated with suicidal behavior in farmers?"
- The figure was included in the discussion (Figure 2). The figure illustrates the associated factors identified in the 14 studies included in this review. Furthermore, it presentsthe proportional calculation of these factors from the main observed categories: work, mental health and social and individual aspects.
This manuscript is a resubmission of an earlier submission. The following is a list of the peer review reports and author responses from that submission.
Round 1
Reviewer 1 Report
18 February 2021
Review on the manuscript titled “Factors associated with suicidal behavior in farmers: a systematic review” by Santos EGO. et al, submitted to IJERPH.
Dear Authors,
The authors systematically reviewed the scientific literature regarding factors associated with suicide in farmers. Out of five databases, two independent researchers found 1441 articles, of which four studies met the final criteria. The authors concluded that several factors were associated with suicide such as depression, drought, herbicides, and insecticides, emphasizing on the importance of further studies regarding factors inherent in farmers, which lead to suicidal behavior.
- Page 1, Abstract: Please list all factors of the studies which met the inclusion criteria.
- Page 1, Abstract: “This study points …”; Please rephrase it to make results and conclusion clear.
- Page 2, Introduction: Please present the rationale to conduct this systematic review.
- Pages 3-10, Results: Many studies were excluded in selection stage. Please describe the main causes of the exclusion.
- Page 5, Figure 2: The figure should be presented after Table 1 and described in the section of risk bias in the studies.
- Pages 7-10, Discussion: Please include a table or figure to summarize it.
- Page 11, Conclusion: It is a repetition of Abstract. Please rephrase it make the contents clearer.
The manuscript contains three figures, one table and 44 references. The reviewer recommends including more references for review articles, preferably more than150 but at least 100. The manuscript carries important value regarding the factors and future tasks associated with suicidal behavior. I reconsider this manuscript for publication after major revision.
I declare no conflict of interest regarding this manuscript.
Best regards,
Reviewer 2 Report
This is a systematic review of suicide in farmers. Here are problems:
1- Only 4 studies are included. This is narrowed from 1400+ articles.
2- It is not well justified in the paper why farmers are at increased risk. The current justification is not enought.
3- The literature review is problematic in my view. They have used "“suicide attempted" so suicide attempt or suicide attempts are left out. Suicide is included but suicidality and suicidal are left out. This problem contrinues. Theses are just a few examples.
4- Any study of suicidality across occupations are left out. A study may say occupation type and suicide and not included here!
Round 2
Reviewer 1 Report
16 March 2021
2nd Review on the manuscript titled “Factors associated with suicidal behavior in farmers: a systematic review” by Santos EGO. et al, submitted to IJERPH.
Dear Authors,
The authors systematically reviewed the scientific literature regarding factors associated with suicide in farmers. Out of five databases, two independent researchers found 1441 articles, of which four studies met the final criteria. The authors concluded that several factors were associated with suicide such as depression, drought, herbicides, and insecticides, emphasizing on the importance of further studies regarding factors inherent in farmers, which lead to suicidal behavior.
- Pages 7-10, Discussion: Please include a figure to present the significance of the study. Informative figures are particularly important for systematic review or meta-analysis, which help the readers understand the article. It can be like a graphical abstract.
- Page 11, Conclusion: Please refine and expand the conclusion.
The manuscript contains three figures, two table and 47 references. The manuscript was partially revised. The reviewer recommends including more references for review articles, preferably more than150 but at least 100. The manuscript carries important value regarding the factors and future tasks associated with suicidal behavior. I reconsider this manuscript for publication after major revision.
I declare no conflict of interest regarding this manuscript.
Best regards,
Reviewer 2 Report
revisions are enough! congratulations!